# START (STrAtegies for RelaTives) coping strategy for family carers of adults with dementia: qualitative study of participants' views about the intervention

Andrew Sommerlad, Monica Manela, Claudia Cooper, Penny Rapaport, Gill Livingston

Division of Psychiatry, University College London, London, UK

**Correspondence to**
Dr Andrew Sommerlad;
a.sommerlad@ucl.ac.uk

## ABSTRACT

**Objectives:** To analyse the experience of individual family carers of people with dementia who received a manual-based coping strategy programme (STrAtegies for RelaTives, START), demonstrated in a randomised-controlled trial to reduce affective symptoms.

**Design:** A qualitative study using self-completed questionnaires exploring the experience of the START intervention. Two researchers transcribed, coded and analysed completed questionnaires thematically.

**Setting:** Three mental health and one neurology dementia clinic in South East England.

**Participants:** Participants were primary family carers of a patient diagnosed with dementia who provided support at least weekly to their relative. We invited those in the treatment group remaining in the START study at 2 years postrandomisation (n=132) to participate. 75 people, comprising a maximum variation sample, responded.

**Primary and secondary outcome measures:** (1) Important aspects of the therapy. (2) Continued use of the intervention after the end of the therapy. (3) Unhelpful aspects of the therapy and suggestions for improvement. (4) Appropriate time for intervention delivery.

**Results:** Carers identified several different components as important: relaxation techniques, education about dementia, strategies to help manage the behaviour of the person with dementia, contact with the therapist and changing unhelpful thoughts. Two-thirds of the participants reported that they continue to use the intervention's techniques at 2-year follow up. Few participants suggested changes to the intervention content, but some wanted more sessions and others wanted the involvement of more family members. Most were happy with receiving the intervention shortly after diagnosis, although some relatives of people with moderate dementia thought it should have been delivered at an earlier stage.

**Conclusions:** Participants' varied responses about which aspects of START were helpful suggest that a multicomponent intervention is suited to the differing circumstances of dementia carers, providing a range of

## Strengths and limitations of this study

- This is the first study to analyse dementia family carers' experiences of and opinions about a clinically effective complex psychological intervention (START, STrAtegies for RelaTives).
- We gained a maximum variation sample; the participants in our study covered the spectrum of sociodemographic and clinical characteristics of the broader group of individuals who had received the START intervention.
- The structured questionnaires were administered 2 years after study entry and response was obtained from 57% of the carers we contacted.
- The use of written questionnaires disadvantaged those with lower literacy and meant we could not probe further. They removed the need to please an interviewer.
- Despite efforts to do so, we were unable to obtain views from carers who had withdrawn from the study, suggesting that we undersampled those who disliked the intervention or found it unacceptable.

potentially helpful strategies. The continued use of the strategies 2 years after receiving the intervention could be a mechanism for the intervention remaining effective.

## INTRODUCTION

The number of people with dementia is increasing due to an expanding older population. There are an estimated 670 000 people in the UK acting as primary family carers for people with dementia, saving the state £8 billion/year.[1] Dementia carers show high levels of psychological distress, including depression and anxiety.[2] This increases

the risk of care home admission for the person with dementia.[3]

Varying interventions have been proposed to support dementia carers, but few are evidence based. Cognitive behavioural therapy reduces carer burden and depression,[4] but it is usually delivered by clinical psychologists who remain a relatively limited resource because they are highly trained and, as a corollary, more expensive. The UK national strategy for improving access to psychological therapies is a stepped care approach, where graduates supervised by clinical psychologists deliver less intensive therapy, allowing clinical psychologists to offer more high intensity interventions to those with more complex needs.[5]

The *Coping with Caregiving* complex psychological intervention was developed in the USA for groups of family carers. It reduced carer depression and anger and improved self-efficacy.[6] We adapted the programme for delivery within the UK National Health Service and evaluated it in the START (STrAtegies for RelaTives) study, a pragmatic randomised controlled trial (RCT). Affective symptoms and case-level depression decreased and quality of life increased in carers receiving the intervention compared to those receiving treatment as usual over an 8-month follow-up period[7] and was cost-effective.[8]

Complex interventions comprise numerous components, which may be independent or interdependent, and the 'active ingredient' is usually hard to determine.[9] Finding out why multicomponent interventions work is important for implementation and might enable the intervention to be refined, tailored for specific groups or reduced in length with associated economic benefits. Researchers have sought to understand mechanisms of action of psychological therapies through exploration of mediators and moderators, such as self-efficacy and coping.[10 11] This is useful but, in addition, participants often have views on which aspects of an intervention were valuable and asking them about this directly, as part of the trial process evaluation, has proved useful in diverse interventions. These have included breastfeeding support interventions,[12] CBT self-management of IBS[13] and maintaining healthy behaviour change.[14]

This approach has not, to the best of our knowledge, been used previously to evaluate complex interventions for dementia carers. We qualitatively analysed dementia carers' experiences of taking part in START, a complex intervention. We aimed to explore which aspects of the therapy carers found helpful and unhelpful; carers' perspectives on the stage of the illness at which the programme should be delivered and how the intervention could be developed to better meet their needs.

## METHODS
### Setting and intervention
The START study was a pragmatic multicentre RCT evaluating the effect on dementia carers' affective symptoms of eight 1 h sessions of a manual-based coping intervention compared to usual treatment. The study protocol has been detailed elsewhere.[7] The intervention was delivered by psychology graduates without clinical qualifications as a face-to-face, individual intervention at a location chosen by the carer, usually their home. The sessions consisted of psychoeducation about dementia, carer stress and access to emotional support; exploration of behaviours or situations that the carer found difficult and potential management strategies; challenging unhelpful thoughts; relaxation techniques accompanied by CDs of relaxation exercises; communication skills; planning pleasurable activities; future planning and maintaining skills learnt. The carers were also given homework to complete and a manual of the intervention in which to record their work. The participants kept the CD and manual to allow their continued use.

### Participants
Consenting participants were included in the main START trial if they identified themselves as the primary family carer of a patient diagnosed with dementia who provided support at least weekly to their relative, who was not living in 24 h care and referred to one of four different settings (three mental health services and a tertiary neurological service for dementia).

In total, 260 carers were randomised, of whom 173 participants were in the intervention group, allocated with a ratio of 2:1 (intervention:treatment as usual) to allow for potential therapist clustering effects in the trial intervention arm. Over the 24-month follow-up period, 41 carers from the intervention group withdrew or were lost to follow-up. We invited the remaining 132 participants to take part in this qualitative substudy.

### Data collection and procedure
At the 24-month follow-up interview, researchers gave participants a questionnaire, a covering letter and a stamped envelope addressed to the trial manager (rather than the researcher with whom they had previously had contact).

The questionnaire was developed with the carers on the trial management and steering committees and consisted of a self-completed questionnaire comprising the following questions:
▶ Was there anything that you found particularly helpful?
▶ How have you used the intervention (support sessions, manual or CD) since it ended?
▶ Is there anything you would do differently?
▶ Is there anything you would add in?
▶ Looking back, do you feel that you took part in the intervention at the right time?

We subsequently sent all participants a transcript of their original response along with a freepost envelope, asking them whether it was representative of their true views and to make amendments if they wished. This method of quality control and validation allows

participants to ensure that the transcript is what they intended to say. We also sent questionnaires to the participants who had previously withdrawn from the study asking the following questions:

▶ What did you think of the support sessions and manual?
▶ Whether you did or did not attend the support sessions, was there anything we should change to make it more useful to you?

We evaluated questionnaire responses alongside sociodemographic and clinical data, including time since diagnosis of dementia, carers' anxiety and depression—measured by the Hospital Anxiety and Depression Scale (HADS),[15] a self-rated scale which has been validated for use in a variety of settings—and the severity of patients' dementia—measured by the clinical dementia rating (CDR),[16] which grades the level of impairment related to dementia. These quantitative data were collected at baseline and at 24 months in the original study.

## Analysis

We transcribed the returned questionnaires verbatim and used a thematic framework approach[17] for analysis. Two researchers (AS and MM) independently read the transcripts and identified a framework of initial themes which referred to the main study objectives. The researchers then used the qualitative software package NVivo (QSR International Pty Ltd, V.9, 2010) to code the transcripts according to these themes and jointly developed a thematic map with a hierarchy of themes and categories. We have anonymised all quotations, providing non-specific demographic information, and do not think that any carer could be identified.

## RESULTS
### Demographics

We received completed questionnaires from 75 participants (57% of the 132 participants at 24 months); 17 of these questionnaires were completed during the research interview with the researcher, who had never been the carer's therapist and the remaining questionnaires were sent by post to our research team. Tables 1 and 2 detail the baseline demographic and clinical characteristics of the participants who received the START intervention and who did and did not complete our questionnaire. Those who did complete the questionnaire covered the demographic and clinical characteristics of the whole group, although spouses or partners of patients were under-represented, and children of people with dementia over-represented; related to this, the mean age of responders was slightly lower in those completing questionnaires and we had fewer responses from retired people and those living with the patient. Comparison using appropriate statistical analysis demonstrates that the lower age of the questionnaire respondents was statistically significant (p=0.03), but the differences in other demographic or clinical characteristics were not statistically significant.

We received only one response from a participant who withdrew; this individual completed the START programme but withdrew from the study before the 24-month follow-up interview. None of the participants who had initially returned a completed questionnaire made notable changes to their responses when invited to do so.

Participants' comments are detailed below and captured within four broad themes: important aspects of the therapy, participants' engagement with the therapy, unhelpful aspects of therapy and potential improvements and appropriate time for delivery of the intervention. Selected quotes are used here to illustrate important viewpoints. We have annotated quotes to describe the participants' role ('w' wife, 'h' husband', 'd' daughter, 's' son, 'n' niece) and numbered participants in the order in which the quotes are used, the severity of dementia at baseline and the carer's total HADS score at the baseline interview and 24-month follow-up (eg, 'HADS 12 → 7'=HADS score of 12 at baseline and 7 at the 24-month interview). The HADS score at 12 months has been provided for two participants who did not complete HADS at 24 months.

### Important aspects of the therapy

Participants valued diverse elements of the intervention and these are summarised in figure 1. The relaxation CDs were most commonly cited as being useful during the period of therapy and beyond, and 22/75 participants told us that they continued to use these and the taught relaxation techniques:

> The CDs are very relaxing … still very much being used today. (w1; very mild dementia; HADS 4 → 13)

> Relaxation exercises helped before bedtime to clear the mind. (d2; moderate dementia; HADS 14 → 10 [12 months])

18 of the 75 participants suggested that understanding the condition in detail made it easier to cope with their relative's symptoms and some mentioned appreciating learning gradually about dementia:

> NHS services gave a lot of information at diagnosis; too much negative info at once. I felt START was more supportive and gave smaller bits at a time. (w3; mild young-onset dementia; HADS 19 → 8)

This knowledge allowed some participants to feel more prepared for the future and this, coupled with effective communication skills, enabled them to cope better as challenges emerged:

> Some of the problems that I eventually had to face had been discussed, making me aware of them and able to care better. (w4; very mild dementia; HADS 12 → 10)

**Table 1** Baseline carer characteristics of questionnaire respondents and non-respondents

| | Respondents (n=75) mean (SD) | Non-respondents (n=98) mean (SD) |
|---|---|---|
| Age | 59.3 (13.7); range: 18–85 | 64.1 (15.1); range: 19–88 |
| Characteristic | n (%) of respondents (n=75) | n (%) of non-respondents (n=98) |
| Gender | | |
| Female | 49 (65.3) | 67 (68.4) |
| Ethnicity | | |
| White UK | 58 (78.4) | 67 (68.4) |
| White other | 4 (5.4) | 12 (12.2) |
| Black and minority ethnic | 12 (16.2) | 19 (19.4) |
| Missing | 1 | 0 |
| Marital status | | |
| Married/common law | 42 (56.0) | 63 (64.3) |
| Education | | |
| No qualifications | 14 (18.7) | 31 (31.6) |
| School level | 24 (32.0) | 27 (27.6) |
| Further education | 23 (30.7) | 24 (24.5) |
| Other | 14 (18.7) | 16 (16.3) |
| Employment | | |
| Full time | 17 (22.7) | 19 (19.4) |
| Part time | 17 (22.7) | 10 (10.2) |
| Retired | 29 (38.7) | 51 (52.0) |
| Not working | 12 (16.0) | 18 (18.4) |
| Relationship to patient | | |
| Spouse/partner | 31 (41.3) | 47 (48.0) |
| Child | 34 (45.3) | 37 (37.8) |
| Other | 10 (13.3) | 14 (14.3) |
| Living with patient? | | |
| Yes | 44 (58.7) | 69 (70.4) |

When she was in hospital, doctors took her off medications. I learnt to be more assertive to talk to doctors and got medications put back on. (s5; talking about Acetyl Cholinesterase Inhibitors; mild dementia; HADS 11 → 7)

Advice on coping with behaviour and communication was cited by 11/75 participants as welcome and was noted by some to have reduced their own distress:

The most important and useful message was to go along with whatever the Alzheimer's sufferer says, i.e. enter their World and don't attempt to correct obvious inconsistencies. (s6; moderate dementia; HADS 7 → 5)

Sessions on carer stress, using a cognitive therapeutic approach to help carers understand their own emotional responses and reframe negative thoughts, were noted by 5/75 participants to have been of practical help; some were grateful for what they saw as a rare chance to explore their own emotional state:

Changing unhelpful thoughts … it concentrated my thoughts on how I was managing my own reactions and trying to be understanding of my husband's illness. (w7; mild young-onset dementia; HADS 14 → 14)

What was an added bonus was that it centred on me rather than my husband. Previously all attention and energy had been focused on them. (w8; moderate dementia; HADS 8 → 11)

17 of the 75 participants told us that they valued the interaction with the therapist for varied reasons. Some were grateful for the opportunity to share their concerns with a professional; others appreciated the personal attributes of their therapist, while yet others noted the empathetic approach of the therapist and the validation of their own feelings:

I think I found the 'talking through' with a knowledgeable person the most helpful. (d9; very mild dementia; HADS 15 → 23)

Therapist was lovely, warm. (w10; very mild dementia; HADS 16 → 16)

I felt it OK to be angry, upset, made to feel less guilty. (d11; very mild dementia; HADS 18 → 13)

10 of the 75 participants commented that the START intervention had a prolonged impact on their lives, either because it empowered them to seek help after the therapy or because they had continued to apply some of the techniques and attitudes to other situations and shared them with other people:

**Table 2** Clinical characteristics of questionnaire respondents and non-respondents

| | Respondents (n=75) median (SD) | Non-respondents (n=98) median (SD) |
|---|---|---|
| Number of months since initial diagnosis | 3.5 (19.8); range: 0–96 | 4.0 (17.3); range: 0–108 |
| HADS baseline | 13.6 (6.9) | 13.4 (7.7) |
| HADS 24m | 14.2 (8.1) | 12.9 (8.3) |
| **Characteristic** | **n (%) of respondents (n=75)** | **n (%) of non-respondents (n=98)** |
| CDR BL | | |
| Very mild | 15 (20.0) | 15 (15.3) |
| Mild | 41 (54.7) | 50 (51.0) |
| Moderate | 19 (25.3) | 29 (29.6) |
| Severe | 0 | 2 (2.0) |
| Missing | 0 | 2 (2.0) |
| CDR 24m | | |
| Mild | 21 (30.4) | 15 (31.3) |
| Moderate | 26 (37.7) | 19 (39.6) |
| Severe | 11 (15.9) | 6 (12.5) |
| Care recipient died | 11 (15.9) | 8 (16.7) |
| Missing | 6 | 9 |
| Withdrawn | 0 | 41 |

CDR, clinical dementia rating score; HADS, hospital anxiety and depression score.

I have since joined the Alzheimer's Society, joined a yoga group and occasionally see a cognitive behavioural therapist—all of which were a result of taking part in the START project. (w7; mild young-onset dementia; HADS 14 → 14)

I have used the methods consistently within my working environment and in offering constructive advice and support to friends dealing with stressful situations that arise within their daily lives. (n12; mild dementia; HADS 25 → 13)

**Influences on carer attitude**
- Greater acceptance of diagnosis and situation
- More tolerant to person with dementia
- Validation of own feelings

**Specific components of START intervention**
- CD and relaxation exercises
- Behavioural and communication advice
- Education about dementia

**Interaction with therapist**
- Sharing concerns with an independent person
- Empathy from therapist
- Reframing of negative automatic thoughts

**Continued use of techniques and attitudes**
- Seeking similar approaches through voluntary organisations
- Sharing techniques with friends and relatives

**Figure 1** Aspects of START (STrAtegies for RelaTives) intervention which were frequently described as helpful by participants.

### Participants' engagement with the therapy

In total, 50 of the 75 participants of those who responded to the questionnaire said that they had continued to use the intervention since the end of the sessions.

> Sometimes I sit and go through my orange folder [therapy manual] and there is a peace and understanding that someone is there with me. (w13; mild dementia; HADS 23 → 17)

Of those who said they had not, 10 gave no reason, 3 said that they had forgotten the sessions and in 2 cases their relative had died during the study. Other stated reasons are described below.

Feeling too busy or tired to continue to engage with the therapy was a frequently cited reason for not continuing to use the intervention, with one participant, the daughter of a woman with Alzheimer's disease, commenting that she had little time to put the strategies into action once the protected therapy time had finished:

> I found it helpful while the sessions were in progress, but lost the allocated time when it was over. (d14; mild dementia; HADS 7 → 5)

Another carer stopped using the intervention because they felt they needed the support and guidance of the therapist. Some respondents commented that they had felt that the intervention was not relevant to their particular situation, either because the dementia was not severe, the caring difficulties did not relate directly to the effect of dementia or because of the particular symptoms they encountered:

> Not really had to use it as my mother is still at an early stage. (s15; mild dementia; HADS 3 → 2)

> Caring problems were mainly physical rather than psychological. (s16; moderate dementia; HADS 9 → 12)

> I felt it was aimed at living with someone who has Alzheimer's which did not apply to me. (d17; very mild dementia; HADS 8 → 14)

Three carers commented that the experience of the START intervention had encouraged them to make use of other techniques:

> Rather than using the CD, I went back to practising transcendental meditation again—so thank you for that. (w8; moderate dementia; HADS 8 → 11)

### Unhelpful aspects of therapy and potential improvements

Eleven of the 75 respondents suggested changes to the START therapy. Some commented that the nature of the intervention did not fit in with their approach or personality:

Wasn't something I would do for myself. (w10; very mild dementia; HADS 16 → 16)

Five of the 75 participants said they would have liked more sessions, with some suggesting a gradual rather than abrupt end to the programme:

Knowing that there would be a follow-up might have kept it all fresher in my mind for longer and got me into a routine of it all better. (d14; mild dementia; HADS 7 → 5)

In contrast, two participants commented that the sessions had been too demanding on their time:

The sessions were too long and interrupted normal daily duties. (w18; mild dementia; HADS 16 → 34)

Five participants suggested that support from other carers through group sessions or attending existing voluntary organisations would have been helpful:

[The Alzheimer's Society café] could have been used as the basis of a carer's group which would be of both practical and emotional help. (w19; mild dementia; HADS 16 → 20)

It was intended that support sessions would include only the therapist and carer, but two participants suggested that other family members could have been included so they too could share the strategies:

Probably add one or two members of family on this programme in case the appointed carer is not able to do the caring. (w1; very mild dementia; HADS 4 → 13)

Although one participant commented that they found it difficult to find a private place for the sessions in the house they shared with the person with dementia, two other carers suggested that including the relative in one or more sessions would have been helpful so that the therapist could tailor the sessions more appropriately or so that the person with dementia could understand the carers' strain:

It would have been nice if the therapist met my Dad … to have the therapist's viewpoint, to see for themselves. (d2; moderate dementia; HADS 14 → 10 [12 months])

One session involving the care-recipient so they appreciate there are problems … and the effect their illness is having on spouse … might help with their self-control. (w10; very mild dementia; HADS 16 → 16)

Two participants stated that the START sessions should have been more explicit in their exploration of the dementia future problems and prognosis:

More discussion of the likely course of the illness. (s20; mild dementia; HADS 17 → 14)

How to prepare for what lies ahead. (h21; moderate dementia; HADS 9 → 26)

Although the CD of relaxation techniques was popular with many respondents, others did not like it:

I haven't used the CD—some of which I found really irritating! (w22; moderate dementia; HADS 22 → 24)

I found the male voices off-putting on the CD—prefer all female voices. (w3; mild young-onset dementia; HADS 19 → 8)

### Appropriate time for delivery of intervention

Participants were largely recruited shortly after or at the time of dementia diagnosis. 61 of the 75 carers judged that they had taken part in the START intervention at the 'right time':

I now feel I have all the tools before she gets worse. (s23; very mild dementia; HADS 12 → 5)

Of those who thought the intervention should have been offered at another time, eight wanted it earlier and six later. About three-quarters of carers looking after those with very mild dementia (CDR 0.5) thought the intervention was delivered at the right time, rising to over 80% of those with relatives with mild or moderate dementia (CDR 1–2). Those who wanted it later tended to have relatives with milder dementia than those asking for it earlier. Among carers who would rather have received the intervention earlier, the median time since they reported being told the diagnosis was 5.5 months, 4 months for those who were happy with the time of delivery and 1.5 months for those who would rather have received it later.

Respondents commented that earlier engagement with the START programme would have helped them improve their communication and thus care better or avoid making major decisions regarding social care without being equipped with the necessary knowledge of dementia:

I wish I knew more, well before her condition was diagnosed, as I feel that I would have been more understanding and giving to her. (d24; mild dementia; HADS 5 → 6)

[START programme] should have started earlier before we found a live-in carer for my mother-in-law. (d25; mild dementia; HADS 11 → 6 [12 months])

Those who felt that the intervention was delivered too early felt it would have helped them cope with their relative's later deterioration:

I feel it was a little early as further down the line, I find it so much harder to cope with my mother as her Alzheimer's has got worse. (d17; very mild dementia; HADS 8 → 14)

## DISCUSSION

This study is the first to qualitatively analyse dementia carers' experiences of a complex psychological intervention and thus help us to understand the mechanisms by which it is effective. We asked carers about their experiences of the therapy 2 years after the 8–12-week therapy began, so we were able to explore how it worked immediately and whether benefits were noted some time later.

Our main finding is that the study participants valued diverse components of the intervention. The most frequently cited aspects were relaxation techniques, education about dementia, interaction with the therapist, cognitive techniques for their own thoughts and feelings, specific strategies for behavioural management and communication techniques for the person with dementia as well as learning to seek and ask for help. Most said that they continued to use techniques at 2-year follow-up.

The heterogeneity of responses suggests that there were no particular aspects that were commonly unnecessary or unhelpful, and that a benefit of the multicomponent nature of this therapy is that it provides a diverse menu of strategies to suit the differing circumstances of dementia carers—in terms of the relative's particular symptoms of dementia; carer knowledge; social situation and support from mental health and social services; preferences and coping strategies.

The HADS scores for each quoted participant allow speculation regarding the impact of their anxiety and depression levels on their mental state at the time of both the intervention and providing this feedback, which is instructive in some cases. However, they are used in illustrative quotes of participants, and cannot be assumed to represent other participants holding a similar view of the intervention.

Some of the responses demonstrate that participants had incorrectly interpreted the intervention's messages—for example, that teaching people with dementia about their condition 'might help with their self-control' or that one should 'go along with whatever the Alzheimer's sufferer says' (the message was to avoid criticism)—reflecting the subjective experience of psychological interventions, but the overall effect of the intervention was nonetheless positive. While some HADS scores went up, for example, the first quote from 'w1', she continued to use the relaxation techniques and felt this was beneficial; it may be that she would have felt worse without the intervention.

Suggestions about how to improve the intervention focused on form rather than content—for example, shorter or more numerous sessions, with several carers suggesting a longer period of intervention with a reduced frequency of sessions. These participants may have been experiencing a higher level of psychological distress, in which case additional sessions for those who are most distressed at the end of therapy would be in line with the stepped care approach to access to psychological therapies. Alternatively, they may have benefited most and so were reluctant to stop.

Family carers have previously reported that receiving information over a longer period helps when making decisions about care,[18–20] and respondents welcomed the gradual accumulation of knowledge about dementia. Others made suggestions about broadening session attendance to include other family members or the person with dementia. While including more family members could reduce the individual feeling of therapist attention, it could broaden the impact of the intervention, and family interventions have been found to be effective in other studies.[21] Perhaps the diverse content helped the START intervention to support carers with a broad range of needs, and a flexible approach to its delivery, in terms of who is present in sessions and how they are scheduled, could assist implementation.

The contact with a professional was welcomed by many participants, who valued the empathetic approach, knowledge and interpersonal skills of the therapists. We know, from an analysis of the effect of clustering by therapists, that the clinical effectiveness of the therapy was not dependent on which therapist delivered the intervention,[7] so this suggests that supervised psychology graduates can deliver this therapy while maintaining a personal approach. Some carers cited a cognitive therapeutic approach as helpful and this supports research findings that cognitive reframing may be an effective aspect of individualised multicomponent interventions.[22]

### Strengths and weaknesses

To the best of our knowledge, our qualitative analysis of participants' experience of a clinically effective and cost-effective psychosocial intervention aimed at improving the mental health of dementia carers is the first study of this type. In order to maximise the validity of our findings, we aimed for and succeeded in gaining a maximum variation sample of people who completed the intervention; the participants in our study covered the spectrum of sociodemographic and clinical characteristics of a broader group of individuals who received the intervention. However, the questionnaire respondents, compared to those who did not respond, were statistically significantly younger and tended to be children rather than spouses of people with dementia, less likely to be married, more likely to be in employment rather than retired and less likely to be living with the person with dementia.

In addition to this, the respondents had reached a higher educational level than non-respondents. It may be that participants with lower literacy attainment would have had more difficulties in filling in the questionnaire. The written format also meant that we could not probe participants' answers. For example, 18 participants specified that they appreciated receiving information about dementia, but we do not know the opinion of the remaining 57 participants about this. Using self-completed questionnaires, however, had the strength that the participants were free to express their views. The lack of changes after we offered participants a

chance to revise their transcripts also suggests this. It also supports the idea that the START intervention had a long-lasting and consistent effect on participants: the initial questionnaire responses providing a snapshot of the participants' views but these remaining constant.

There is probably some response bias, with those who valued and benefited from the therapy most or least and had the strongest feelings being more likely to respond. As we did not receive any responses from participants whose relative had severe dementia at the beginning of the intervention, we cannot make assumptions about the experience of the intervention for this group. Nonetheless, many of the respondents cared for people who progressed to severe dementia or died, so delivering the intervention early may mean that it continues to confer benefit as dementia deteriorates. We have been unable to obtain views from most carers who withdrew from the study, which would suggest that we probably undersampled those who disliked the intervention or found it unacceptable. Nonetheless, a minority of respondents did criticise the therapy, suggesting that our strategies to minimise social desirability bias were successful.

### Clinical implications and future research

This analysis indicates aspects of the START intervention which were helpful and suggests some possible reasons for lasting clinical efficacy.[23] Participants took part in a structured and guided intervention and their comments indicated a range of opinions about which parts were helpful. It seems that, although some individuals did not like components of the intervention, most were able to identify helpful aspects and that, by working with graduate psychologists over a period of time, they made some longer term changes. Supporting the structured sessions with a manual and CD-based intervention was valuable for the two-thirds of participants who continued to use them 2 years later, and participants commented that they also felt more capable of seeking help for future changing circumstances.

Our analysis has implications for when it is best to offer the intervention to carers in order to maximise engagement and potential benefit. Responses suggest that most carers want the intervention to be delivered shortly after they had been told of a dementia diagnosis; this allows future social care planning and psychological preparation for the developing illness. Some carers for people with very mild dementia would have preferred to wait a few months following diagnosis. In clinical practice, having flexibility in terms of when people are offered the START intervention could increase the acceptability of the intervention.

Overall, we found that two-thirds of carers reported continued use of the intervention after 2 years. This indicates a possible mechanism for continuing efficacy after the end of the intervention. Retaining a copy of the manual and CD provides an opportunity to revisit valued aspects of the therapy and adapt caring

approaches to current challenges, while signposting to relevant voluntary organisations has been shown to be helpful[24] and empowered the participants in our study to seek help elsewhere, providing help as circumstances changed. Our findings are in line with others that multi-component interventions are necessary in complex conditions with multiple domain pathology and that a single active ingredient may be illusory.[25] [26]

**Acknowledgements** The authors thank Dolores Gallagher Thompson for her original manual and allowing us to adapt it; the participating carers; the Camden and Islington NHS Foundation Trust, University College London Hospital, the North East London Foundation Trust, and the North Essex Partnership Foundation; Vincent Kirchner and Lisa Gee for referring many patients; members of the steering committee and the data monitoring committee. The START research team acknowledges the support of the National Institute for Health Research, through the Dementia and Neurodegenerative Research Network (DeNDRoN).

**Contributors** AS, PR, CC and GL contributed to the design of the study. MM facilitated the questionnaire distribution. AS and MM analysed the data for themes and agreed on a coding frame. All authors contributed substantially to the conception and design, or analysis and interpretation of data, and to drafting the article or revising it critically for intellectual content.

**Funding** This project was funded by the National Institute for Health Research Health Technology Assessment (HTA) programme (project no 08/14/06). The study was sponsored by University College London.

**Competing interests** None.

**Ethics approval** The trial was conducted in accordance with Good Clinical Practice guidelines, the Declaration of Helsinki, the Clinical Trials Regulations and local laws and regulations. We obtained written ethics approval for the study from East London and the City Research Ethics Committee for the trial (ID: 09\H0703\84) and Research and Development permission from the local trusts.

**Provenance and peer review** Not commissioned; externally peer reviewed.

**Data sharing statement** No additional data are available.

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
