## [Reviewer comments · BMJ Open]

Some articles will have been accepted based in part or entirely on reviews undertaken for other BMJ Group journals. These will be reproduced where possible.

ARTICLE DETAILS

TITLE (PROVISIONAL)	START (STrAtegies for RelaTives) coping strategy for family carers of adults with dementia: Qualitative study of participants' views about the intervention.
AUTHORS	Sommerlad, Andrew; Manela, Monica; Cooper, Claudia; Rapaport, Penny; Livingston, Gill

VERSION 1 - REVIEW

REVIEWER	Edward Strivens Cairns and Hinterland Hospital and Health Service, Australia James Cook University School of Medicine, Australia
REVIEW RETURNED	25-Apr-2014

GENERAL COMMENTS	In presenting the demographics, results and comments from carers, I found the format of the free text made it more difficult to get to the crux of the issues. The paper may benefit from both the demographics and exemplar comments being presented in tabular form. The main limitation of the study probably involves the use of questionnaires rather than a semi-structured interview. This is acknowledged in the paper. It does impact on the ability to 'unpack' the interventions further, especially with regard to content versus form of the program. Overall an interesting and novel evaluation of participants views on this encouraging program of complex intervention. As you identified in the paper, a semi structured interview technique may had added to the information gathered and may also have helped with the under-representation of partners and spouses.
---

REVIEWER	Karen B. Hirschman, PhD University of Pennsylvania School of Nursing Philadelphia, Pennsylvania U.S.A.
REVIEW RETURNED	01-May-2014

GENERAL COMMENTS	Thank you for the opportunity to review this interesting qualitative analysis of carers' experience with the START program in the UK. Overall these findings shed new light from the perspective of the carers' who participated in the START program. The authors do acknowledge that the findings may be biased as only those START
---

	participants who did not withdraw from the study responded to the questionnaire. See additional comments below aimed at strengthening this manuscript. Page 4 1) Minor editing: the sentence at the top of page 4 starting with “Affective symptom and case-level...” remove the word “this” at the end (line 6). 2) Please add a statement or statements about the larger study’s ethics review (e.g., institutional review board approval?) It is not enough to state that consenting participants were included. (line 57) Page 5 3) How many surveys were returned by mail? I suggest moving the detail about the 17 cases in which the subject completed the survey during the 24m interview be moved to the results. OR add in here the number returned by post. Page 6 4) Please add references to the Analysis paragraph (lines 5-13). There should be qualitative data analyses references for your thematic framework approach. 5) Results – were any of the differences between the two groups (returned survey group vs. non-completers)? Can you run some basic comparisons? Chi2, ttest (or more conservative non-parametric test)? 6) Please add a small paragraph (2-3 sentences) before launching into the themes. This paragraph should provide enough information that the reader knows where this section is heading with the specific themes. Pages 6-11 7) For consistency and perspective, it would be very helpful if the n/N was consistently noted throughout. In some places the N (%) is listed or just a % or just an N [examples: 11/75; 15% (11/75)]. While some of the themes appear to be endorsed by only a few caregivers some themes are endorsed by the vast majority. Page 14 8) Lines 14-15 : Please add a reference or references for this statement about the “lasting clinical efficacy”.
--	---

VERSION 1 – AUTHOR RESPONSE

Reviewer 1: Edward Strivens

In presenting the demographics, results and comments from carers, I found the format of the free text made it more difficult to get to the crux of the issues.

The paper may benefit from both the demographics and exemplar comments being presented in tabular form.

The main limitation of the study probably involves the use of questionnaires rather than a semi-structured interview. This is acknowledged in the paper. It does impact on the ability to 'unpack' the interventions further, especially with regard to content versus form of the program.

Overall an interesting and novel evaluation of participants views on this encouraging program of complex intervention. As you identified in the paper, a semi structured interview technique may have added to the information gathered and may also have helped with the under-representation of partners and spouses.

RESPONSE:

Thank you for your comments on our paper – we are glad that you found it an interesting analysis. We are keen to keep the free-text format in order to make the results section more readable, but have made some changes to the presentation of the results, adding an introductory paragraph and more

consistent information on the number of participants sharing particular viewpoints And we feel that this makes it easier for the reader to extract important information. We would, however, be happy to put these results into table format, should the editors wish for us to do so.

Reviewer 2: Karen B. Hirschman

Thank you for the opportunity to review this interesting qualitative analysis of carers' experience with the START program in the UK. Overall these findings shed new light from the perspective of the carers' who participated in the START program. The authors do acknowledge that the findings may be biased as only those START participants who did not withdraw from the study responded to the questionnaire. See additional comments below aimed at strengthening this manuscript.

1) Minor editing: the sentence at the top of page 4 starting with "Affective symptom and case-level..." remove the word "this" at the end (line 6).

2) Please add a statement or statements about the larger study's ethics review (e.g., institutional review board approval?) It is not enough to state that consenting participants were included. (line 57)

3) How many surveys were returned by mail? I suggest moving the detail about the 17 cases in which the subject completed the survey during the 24m interview be moved to the results. OR add in here the number returned by post.

4) Please add references to the Analysis paragraph (lines 5-13). There should be qualitative data analyses references for your thematic framework approach.

5) Results – were any of the differences between the two groups (returned survey group vs. non-completers)? Can you run some basic comparisons? Chi2, ttest (or more conservative non-parametric test)?

6) Please add a small paragraph (2-3 sentences) before launching into the themes. This paragraph should provide enough information that the reader knows where this section is heading with the specific themes.

7) For consistency and perspective, it would be very helpful if the n/N was consistently noted throughout. In some places the N (%) is listed or just a % or just an N [examples: 11/75; 15% (11/75)]. While some of the themes appear to be endorsed by only a few caregivers some themes are endorsed by the vast majority.

8) Lines 14-15 : Please add a reference or references for this statement about the "lasting clinical efficacy".

RESPONSE:

Thank you for your interest and helpful comments on our paper:

1) We have made this change as suggested.

2) We have added a section on the main study's ethical approval at the end of our manuscript.

3) As suggested, we have added this information to the results section of the paper and explained that, of the 75 completed responses, 17 were obtained during interview.

4) Appropriate reference added.

5) We have run these analyses and have added a comment on this in the main paper. All comparative results were non-significant, except for differences in age between the groups (reflecting, as mentioned, the higher rate of responses from children rather than spouses.)

6) We have added some introductory information before the main body of results, detailing the way in which these will be structured according to the main themes.

7) We have amended the results in order that the numbering is more consistent as you suggested.

We have included results as, for example, 12/75 where relevant and removed the percentages where previously used. We agree that this adds perspective for the reader although, as explored in the discussion section, the written format of the questionnaires means that these numbers are not an entirely accurate reflection of the participants' views.

8) We have added a reference for this, reflecting the favourable results for the START intervention at 24 month follow-up in order to justify the statement on 'lasting clinical efficacy'.